# Association between Meteorological Factors and Mumps and Models for Prediction in Chongqing, China

**DOI:** 10.3390/ijerph19116625

**Published:** 2022-05-29

**Authors:** Hong Zhang, Kun Su, Xiaoni Zhong

**Affiliations:** 1School of Public Health and Management, Chongqing Medical University, Chongqing 400016, China; 2020111437@stu.cqmu.edu.cn (H.Z.); sukun325@163.com (K.S.); 2Chongqing Municipal Center for Disease Control and Prevention, Chongqing 400042, China; 3Chongqing Public Health Medical Center, Chongqing 400036, China

**Keywords:** mumps, ARIMA model, multivariate time series analysis

## Abstract

(1) Background: To explore whether meteorological factors have an impact on the prevalence of mumps, and to make a short–term prediction of the case number of mumps in Chongqing. (2) Methods: K–means clustering algorithm was used to divide the monthly mumps cases of each year into the high and low case number clusters, and Student *t*–test was applied for difference analysis. The cross–correlation function (CCF) was used to evaluate the correlation between the meteorological factors and mumps, and an ARIMAX model was constructed by additionally incorporating meteorological factors as exogenous variables in the ARIMA model, and a short–term prediction was conducted for mumps in Chongqing, evaluated by MAE, RMSE. (3) Results: All the meteorological factors were significantly different (*p* < 0.05), except for the relative humidity between the high and low case number clusters. The CCF and ARIMAX model showed that monthly precipitation, temperature, relative humidity and wind velocity were associated with mumps, and there were significant lag effects. The ARIMAX model could accurately predict mumps in the short term, and the prediction errors (MAE, RMSE) were lower than those of the ARIMA model. (4) Conclusions: Meteorological factors can affect the occurrence of mumps, and the ARIMAX model can effectively predict the incidence trend of mumps in Chongqing, which can provide an early warning for relevant departments.

## 1. Introduction

Mumps is an acute respiratory infectious disease caused by the mumps virus, which is highly contagious and could be transmitted through direct contact and droplets [1]. Children and adolescents are susceptible to mumps and prone to outbreaks in collective units, such as primary schools, secondary schools and childcare institutions. Although mumps is a category–C infectious disease (category–C infectious diseases are also known as surveillance and management infectious diseases, and such infectious diseases should be managed by the relevant departments) in China, it can easily be accompanied by meningitis, orchitis, oophoritis and other complications, causing considerable harm to children and adolescents [2]. The mumps–containing vaccines have been included in the national routine program for immunization since 2008 in China. Although the implementation of immunization programs played a role in controlling the occurrence of mumps in Xinjiang, Tibet and Ningxia provinces, there was no significant change in the overall incidence rate across the country [3].

Extreme weather events may cause or exacerbate the outbreak and prevalence of infectious diseases [4,5,6]. As for mumps, many statistical methods have been applied to discuss the relationship between mumps and meteorological factors. For example, Zhang, D. constructed a boosted regression–tree model to study the nonlinear and delayed effects between the meteorological factors and mumps [7]. A distributed lag non–linear model (DLNM) was applied in Hu, W.’s study, demonstrating that the upper level of precipitation, atmospheric pressure and relative humidity could increase the risk of mumps [8]. Other studies using the ARIMAX model showed that temperature, wind velocity and atmospheric pressure affected the occurrence of local mumps in Beijing, Taiwan and Guangzhou [9,10,11]. However, areas in those studies were mainly plain. Chongqing, located in the subtropical inland area of the northern hemisphere, was surrounded by plateaus and mountains on all sides. Affected by multiple climate systems, the weather there was complex, and extreme weather events, such as high–temperature events, low–temperature events, and heavy rain events, occurred from time to time, which aggravated the risk of a variety of diseases [12,13,14]. At present, there are a lack of studies on the relationship between the meteorological factors and mumps in Chongqing, so it is crucial to construct the incidence prediction model of mumps in Chongqing based on meteorological factors.

The auto–regressive integrated moving average (ARIMA) model is one of the common time series models and is applied in various areas of short–term prediction [15,16,17,18,19]. However, it cannot explain the effect of exogenous variables, and the prediction accuracy may be limited by obvious influencing factors. Recently, the ARIMAX model was widely applied for infectious disease prediction [20,21,22,23,24]. In our study, the ARIMAX model is proposed to predict the occurrence of mumps in Chongqing based on meteorological factors.

## 2. Materials and Methods

### 2.1. Study Area

Located in the middle and lower reaches of the Yangtze River, Chongqing is the largest industrial and commercial city in southwest China with 76% of the area being mountainous. It contains 38 districts, with an area of 82,400 square kilometers, and a permanent population of more than 30 million (ttps://www.cq.gov.cn/zjcq/ (accessed on 15 May 2022)).

### 2.2. Data Sources

Monthly cases of mumps in Chongqing from 2009 to 2019 were collected from the Chongqing Centers for Disease Control and Prevention. Additionally, meteorological data of Chongqing from 2009 to 2019 were obtained from Chongqing Statistical Yearbook (http://tjj.cq.gov.cn/ (accessed on 13 September 2021)), including monthly precipitation (*p*, mm), average temperature (T, °C), sunshine duration (S, h), relative humidity (H, %), atmospheric pressure (Pr, Pa), wind velocity (W, m/s) and rainy days (R, d).

### 2.3. Data Preprocessing

All mumps cases were verified through clinical and laboratory diagnoses, and reported to the CDC by health departments. Download all the mumps case report cards from 2009 to 2019 in Chongqing, delete the duplicate cards and then construct a time series of monthly cases by the onset date. Download the meteorological data from the 2009 to 2019 Chongqing Statistical Yearbook, and construct time series of the meteorological factors, respectively.

### 2.4. Statistical Analysis

In this study, R version 4.0 and SAS version 9.4 were used for statistical analysis. K–means clustering algorithm was used to divide the monthly mumps cases of each year into high and low groups, and Student’s *t*–test was applied for difference analysis. ARIMAX model was used to predict mumps in Chongqing and compared with the ARIMA model.

The ARIMA model is generally expressed as ARIMA(p,d,q)(P,D,Q)_[n]_. Among them, d, D are the non–seasonal–difference and seasonal–difference orders; p, q are the auto–regressive and moving average orders; and P, Q are the seasonal auto–regressive and moving average orders. n is the period of the sequence. The ARIMAX model is based on ARIMA, X is an exogenous variable and its formula is [25]:$$\{\begin{array}{ll} y_{t}=\mu & +\overset{k}{\underset{i=1}{\sum }}\frac{\Theta_{i}\left(B\right)}{\Phi_{i}\left(B\right)}B^{l_{i}}x_{it}+\varepsilon_{t}\\ & \varepsilon_{t}=\left(\frac{\Theta \left(B\right)}{\Phi \left(B\right)}a_{t}\right) \end{array}$$

In the above equation, $y_{t}$ and $x_{it}$ are the dependent and independent variable sequences, $l_{i}$ is the lag effect; B represents the backward–shift operator, $\Theta_{i}\left(B\right)$ and $\Phi_{i}\left(B\right)$ are the moving average coefficient polynomial and auto–regressive coefficient polynomial of the independent variable; $\varepsilon_{t}$ is the regression residual; Θ(B) and Θ(B) are the moving average coefficient polynomial and auto–regressive coefficient polynomial of residuals; and $a_{t}$ is the white noise sequence with zero mean. The processes of building the ARIMAX model for mumps are as follows:(1)Establish ARIMA model for mumps: Use the monthly reported cases of mumps in Chongqing from January 2009 to December 2018 to establish an ARIMA model. Firstly, the ADF test is used to examine whether the sequence is stationary (*p* < 0.05 indicates non-stationary), and the Box–Ljung test is to determine whether the sequence is a white noise sequence (*p* < 0.05 indicates that the sequence is non-white noise). Secondly, the model orders are measured according to the auto–correlation function plot (ACF) and partial auto–correlation function plot (PACF). Additionally, the least–squares method (LSM) and the Student’s *t*−test are used for parameter estimation and testing (*p* < 0.05 indicates statistically significant parameters), and the Box–Ljung test is conducted for model diagnosis (*p* ≥ 0.05 sufficiently indicates the model extract information). Finally, the Akaike information criterion (AIC) is used to select the optimal model among all ARIMA(p,d,q)(P,D,Q)_[n]_ models that passed the tests (the smaller the AIC, the better the model fitting).(2)Select exogenous variables: Use the cross–correlation function (CCF) plot to assess the relationship between mumps and meteorological factors, and to determine which factor and its lag order are suitable for the ARIMAX model. In the CCF plot, the horizontal axis is the lag order, the vertical axis is the correlation coefficient and the dashed line is the reference line 2 times the standard deviation. If the coefficient at some lag order exceeds 2 standard deviations, it can be considered that the meteorological factor at that lag order is correlated to mumps.(3)Model selection: The exogenous variables obtained in the previous step are incorporated into the ARIMA model to fit the ARIMAX model, with the parameter test and model diagnosis is performed. The best ARIMAX model is selected by AIC from models that have passed parametric tests and model diagnoses.(4)Model prediction: The ARIMA and ARIMAX models are used to predict the monthly case number of mumps in Chongqing in 2019, and compared with the actual cases, respectively. The mean absolute error (MAE) and the root–mean–square error (RMSE) are used to evaluate the prediction error, and the smaller the MAE and RMSE, the smaller the prediction error. The formulas are as follows:$$\{\begin{array}{c} MAE=\frac{\sum_{t=1}^{T}\left|{\hat{y}}_{t}-y_{t}\right|}{T}\\ RMSE=\sqrt{\frac{\sum_{t=1}^{T}{\left({\hat{y}}_{t}-y_{t}\right)}^{2}}{T}} \end{array}$$

In the above equation, ${\hat{y}}_{t}$ is the predicted number of monthly reported cases and $y_{t}$ is the actual number of the monthly reported cases.

## 3. Results

### 3.1. Descriptive Analysis

A total of 127,107 confirmed cases of mumps were reported in Chongqing from 2009 to 2019. The confirmed cases have increased since 2009, then peaked in 2011 with 17,455 confirmed cases and then declined, with the lowest case of 6493 in 2019. The average age of the individuals in these cases was 10 years old, sex ratio was 1.40:1 (male to female) and people under the age of 18 were the mainstay, accounting for 90.76% (see Appendix A).

### 3.2. Difference Analysis

K–means clustering algorithm divided monthly cases of each year into two clusters. The mean of mumps cases in cluster A was 1793.70, significantly larger than that for cluster B (708, *p* < 0.05). Cluster A represented the months with the high number of mumps cases, with a total of 31 months, and cluster B represented the months with the low number of mumps cases, with a total of 101 months. We compared the meteorological factors between clusters A and B and found that almost all the meteorological factors were significantly different (*p* < 0.05), except for relative humidity. The mean of monthly precipitation, temperature, sunshine duration, wind velocity and rainy days for cluster A were significantly higher than those for cluster B, while the monthly average atmospheric pressure for cluster A was lower than that for cluster B (see Table 1, Figure 1).

### 3.3. Model Construct

#### 3.3.1. ARIMA Model

Monthly cases of mumps from 2009 to 2018 were chosen as the training set. The sequence diagram indicated a seasonal tendency, and the peak of the cases gradually decreased as the years passed (see Figure 2). ACF and PACF plots of the series showed that ACF and PACF of the sequence were both trailing (see Figure 3). Considering that there were obvious periodic characteristics and a downward trend of the series, a one–step analysis and a period of 12 seasonal differences were performed to make it stationary. The ADF and Box–Ljung tests manifested that the differential sequence was stationary and non–white noise (see Table 2). ACF and PACF plots of the stationary sequence demonstrated that the ACF and PACF also had the characteristics of tailing, so the ARIMA model could not be directly ordered by the ACF and PACF plots (see Figure 3). Since the values of p, q, P and Q generally did not exceed 2, trial orders from 0 to 2 were performed. All the possible models were established, of which 15 passed the parametric test, and the model ARIMA(1,1,2)(2,1,0)_[12]_ with the minimum AIC was the best (see Table 3).

#### 3.3.2. ARIMAX Model

In this study, CCF plots were used to explore the association between meteorological factors and mumps, and the selected variables obtained in the above steps were incorporated into the ARIMA model to construct the ARIMAX model. Since our study was based on monthly data, only the cross–correlation within 12 months of lag was considered. The results show that all meteorology factors have cross–correlation coefficients exceeding the 2 times of standard deviation, indicating that these meteorological factors may have an influence on mumps. Meanwhile, these influences presented a significant lag effect and differed among different meteorological factors. For instance, the precipitation lags at 3, 4, 5 and 6 months were negatively correlated to mumps, while lags 0, 9, 10, 11 and 12 months were positively correlated to mumps. The relative humidity lags 2, 3, 9 and 10 months were negatively correlated to mumps, while lags 5, 6 and 7 months were positively correlated to mumps (see Figure 4).

The meteorological factors with corresponding lags in the above results were included in ARIMA(1,1,2)(2,1,0)_[12]_, and then the ARIMAX model was obtained. Model test and parameter test results show that the ARIMAX model with the 10–month lag precipitation, 9–month lag temperature, 9–month lag relative humidity and 7–month lag wind velocity passed the parameter tests (*p* < 0.05) and model diagnosis, and the AIC of the ARIMAX model with 10–month lag precipitation was 1328.84, lower than that of the other models, so we chose the ARIMAX model with 10–month lag precipitation as the best ARIMAX model (see Table 4).

#### 3.3.3. Model Prediction

The ARIMAX and ARIMA models obtained above were applied to predict mumps cases in 2019, respectively. Model 1 represented the ARIMA model; model 2 represented the ARIMAX model incorporating 10–month lag precipitation. Comparing model 2 with model 1, the RMSE of model 2 was relatively lower than that of model 1 (RMSE decreased by 4.35%), indicating that the prediction effect of the ARIMAX model was slightly better than the ARIMA model (see Table 5). As is shown in the figure, the ARIMA model accurately predicted the case number in February, March and April, while the predictions in January and May were higher than what we observed. The case numbers predicted by the ARIMAX model were relatively accurate, except for January and February. In general, ARIMAX models could accurately predict the incidence trend of mumps in Chongqing in 2019 (see Figure 5).

## 4. Discussion

Studies have shown that the epidemic of infectious diseases has a clear seasonal trend, but the mechanism may be complex and have not yet been fully understood. Scholars believe that it may be related to changes in the environment and the behavior of susceptible populations, especially for respiratory infectious diseases, for which environmental factors may play important roles in the transmission process [26,27]. Through K–means clustering and difference analysis, we speculated that higher temperature, precipitation, sunshine duration, wind velocity and lower atmospheric pressure might be risk factors for the occurrence of mumps. Additionally, through CCF and ARIMAX models, we finally determined that precipitation, temperature, relative humidity and wind velocity were associated with mumps, and there were obvious lag effects.

The results of the K–means clustering algorithm showed that the high case number cluster was concentrated from April to July for each year, indicating that the peak of mumps was in summer, which was consistent with a previous study conducted in Chongqing [16]. The meteorological feature of the two clusters exhibited a significant difference, and the mean of monthly precipitation, temperature, sunshine duration, wind velocity and rainy days of the high case number cluster were significantly higher than those of the low case number cluster, while the monthly average atmospheric pressure was lower than low case number cluster, suggesting that the meteorological characteristics of high–case–number months in Chongqing could be featured by warm, abundant precipitation, sufficient sunshine, higher wind velocity and lower atmospheric levels.

Temperature can directly affect the state of virus proteins and genes and is the most important factor affecting virus survival. In other cities in China, the temperature is one of the most important factors affecting mumps [8,28,29,30]. A conducted study in Taiwan showed that, when the temperature was 20 °C, the cases of mumps began to increase, and the cases of mumps began to decline at 25 °C [10]. In our study, we found that the average temperature in the months with high case numbers of mumps was 22.5 °C, and 17.7 °C in months with a low case number of mumps, and the average temperature 9–month lag was positively correlated to mumps (β = 20.37, *p* = 0.04). Meanwhile, a study showed that physical activity during adolescence was increased during warmer months [31]. Therefore, it is reasonable to suspect that warmer weather can affect the spread of mumps by increasing people’s contact.

Relative humidity can also affect virus survival. Mumps virus has a lipid envelope and can survive longer in environments with lower relative humidity [27]. A study conducted in Jining found that, when relative humidity with a 14-day lag exceeded 54%, for every 1% increase in relative humidity, the excess risk of mumps increased by 1.86% [32]. In our study, though there was no significant difference in the relative humidity between the high and low case number clusters, the results of the CCF and ARIMAX model indicated the relative humidity 9–month lag was negatively correlated to mumps (β = −6.10, *p* = 0.04), which was mainly due to the high relative humidity all year round in Chongqing.

The effects of other meteorological factors on mumps have also been reported, though the mechanism has not been fully understood. Zha WT’s research found that the 5–month lag precipitation in northern China was negatively correlated with the incidence of mumps [33]. However, in our study, the 10–month lag precipitation was positively correlated with mumps (β = 0.46, *p* = 0.03), and the average precipitation in months with a high case number of mumps was 161.6 mm, significantly higher than that in low–case–number months (81.82 mm, *p* < 0.05). Li’s study indicated that, when the sunshine duration during 1-day lag exceeded 5/h per day, the excess risk of mumps increased by 12.94% for every 1/h increase in the sunshine duration [32]. In our study, we found the mean of sunshine duration in the months with a high case number of mumps was longer than that in low–case–number months. Yang, Q’s research proved that the relative risk of mean wind velocity was 0.70 (95%CI, 0.54 to 0.91) comparing the 1st percentile to the median [11]; in our study, the mean of wind velocity was significantly higher in high–case–number months and the wind velocity lag 7 months was positively correlated to mumps (β = 207.94, *p* = 0.03). It can be observed that the association between meteorological factors and mumps was not consistent in different countries, and we doubt that these differences may be attributed to environmental and climatic characteristics in different regions.

This study also had some limitations. Due to the data type, our model cannot explain the short-term effect of meteorological factors on mumps for the lack of daily or weekly data, and we only incorporated a single meteorological factor into the ARIMAX model. In addition, for the lack of the coverage rate of the mumps–containing vaccines in Chongqing, the impact of the vaccines on mumps was not discussed. Further research regarding the short–term effects of multiple meteorological factors and vaccines on the occurrence of mumps remains to be conducted.

## 5. Conclusions

The results demonstrate that, in Chongqing, certain meteorological factors, such as monthly average precipitation, monthly average temperature, monthly average relative humidity and monthly average wind velocity, have an influence on the monthly case number of mumps, and these influences present a significant lag effect. Additionally, the ARIMAX model incorporating meteorological factors can effectively predict the epidemic trend of mumps in Chongqing. Through the predictive model, the future epidemic trend of mumps can be known in advance, and it can provide an early warning for the relevant departments.

## Figures and Tables

**Figure 1 ijerph-19-06625-f001:**
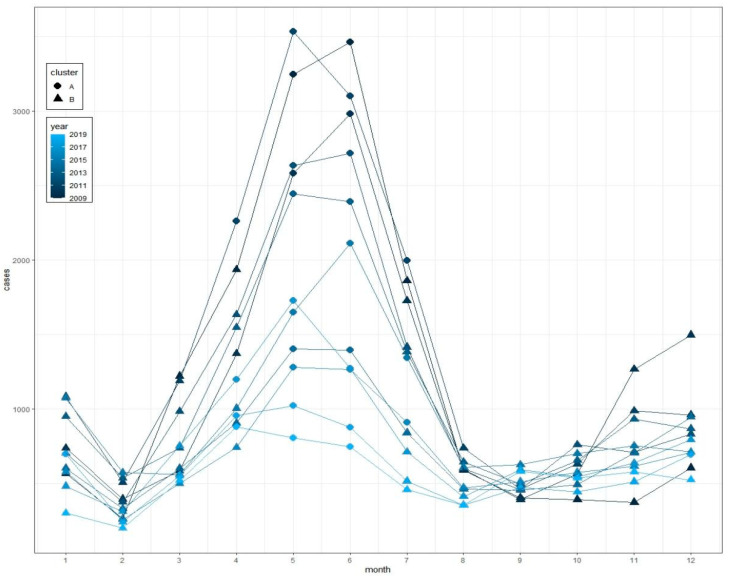
Monthly cases of mumps and clusters from 2009 to 2019 in Chongqing.

**Figure 2 ijerph-19-06625-f002:**
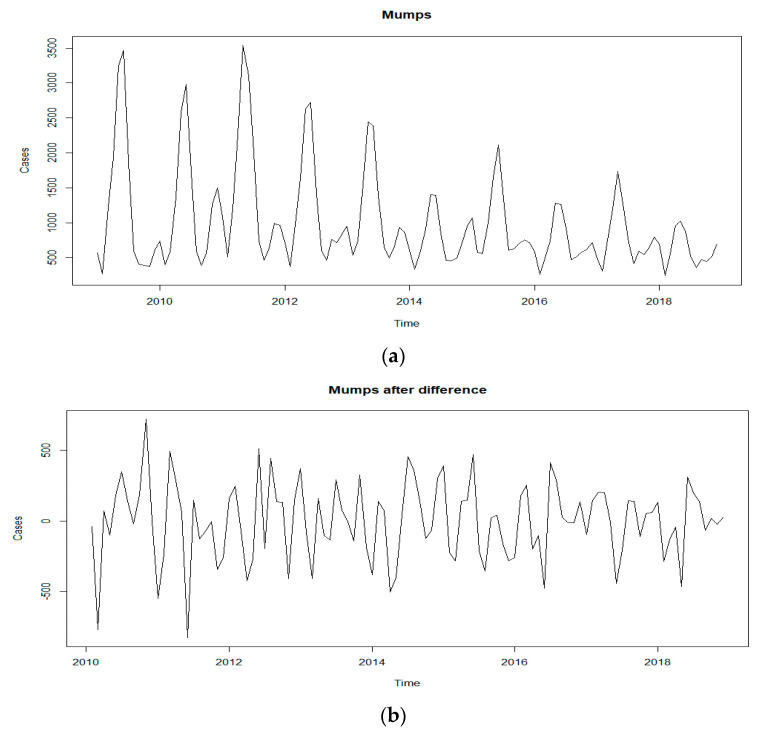
Sequence diagram of mumps: (**a**) Reported monthly mumps cases from January 2009 to December 2019; (**b**) monthly mumps cases after difference.

**Figure 3 ijerph-19-06625-f003:**
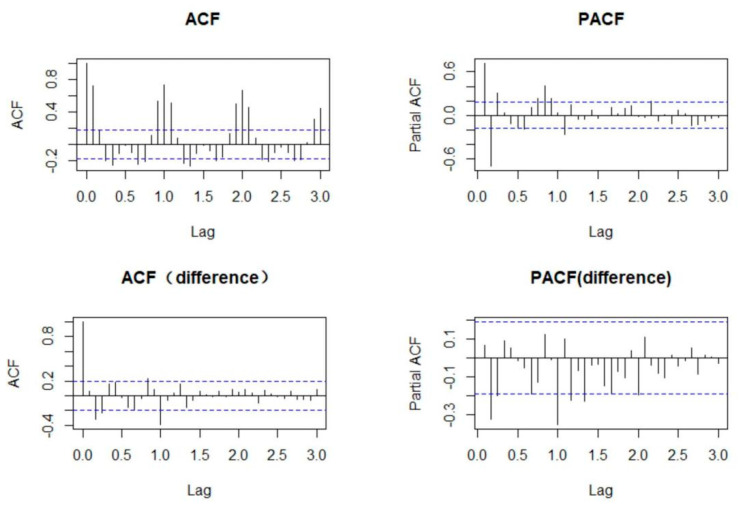
ACF and PACF plots of mumps and difference mumps sequences.

**Figure 4 ijerph-19-06625-f004:**
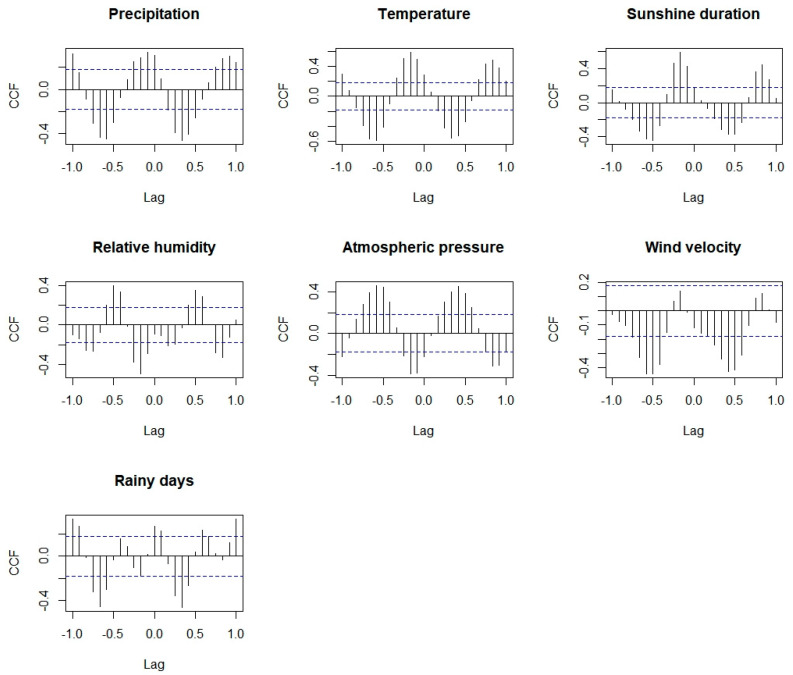
Cross–correlation between mumps and meteorological factors.

**Figure 5 ijerph-19-06625-f005:**
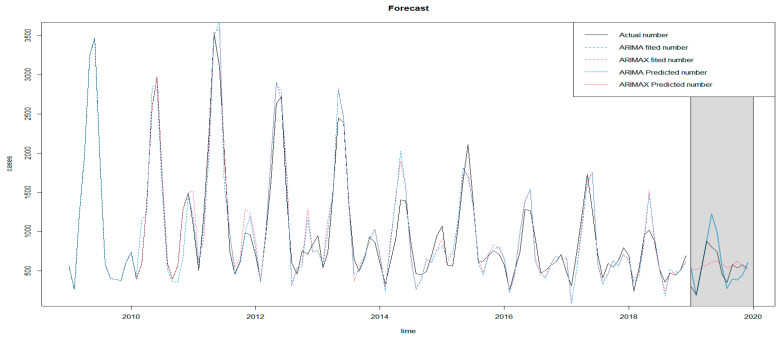
Prediction diagram of ARIMA and ARIMAX.

**Table 1 ijerph-19-06625-t001:** *t*–test of meteorological factors and mumps cases in Chongqing, 2009–2019.

Factor	Cluster	Mean	Std	T–Value	*p*–Value
Mumps	A (N = 31)	1793.70	900.40	9.91	<0.01 *
	B (N = 101)	708.00	356.30		
P	A	161.60	78.68	5.18	<0.01 *
	B	81.82	73.84		
T	A	22.50	4.80	4.43	<0.01 *
	B	17.44	7.53		
S	A	108.20	54.32	2.29	0.02 *
	B	78.93	64.60		
H	A	76.70	5.27	−0.05	0.96
	B	76.77	7.51		
W	A	1.51	0.24	2.06	0.04 *
	B	1.41	0.21		
Pr	A	970.60	6.59	−5.29	<0.01 *
	B	979.70	8.84		
R	A	15.97	4.13	4.42	<0.01 *
	B	12.16	4.23		

* Represents *p* < 0.05.

**Table 2 ijerph-19-06625-t002:** ADF and Box–Ljung tests for mumps after difference.

ADF Test			Box–Ljung Test		
Type	Lag	*p*	χ-Squared	Df	*p*
No drift no trend	0	<0.01	23.62	6	<0.01
	1	<0.01			
	2	<0.01			
With drift no trend	0	<0.01	56.96	12	<0.01
	1	<0.01			
	2	<0.01			
With drift and trend	0	<0.01	65.14	18	<0.01
	1	<0.01			
	2	<0.01			

**Table 3 ijerph-19-06625-t003:** All ARIMA models that passed the parametric test and their AIC.

Model	AIC	Box–Ljung Test *p*
ARIMA(0,1,0)(0,1,1)_[12]_	1490.14	0.13
ARIMA(0,1,0)(1,1,0)_[12]_	1488.78	0.28
ARIMA(1,1,1)(1,1,0)_[12]_	1488.04	0.69
ARIMA(1,1,2)(0,1,0)_[12]_	1494.37	0.99
ARIMA(1,1,2)(0,1,1)_[12]_	1471.39	0.86
ARIMA(1,1,2)(1,1,0)_[12]_	1470.60	0.96
ARIMA(1,1,2)(2,1,0)_[12]_	1468.68 *	0.83
ARIMA(0,1,1)(0,1,1)_[12]_	1487.68	0.61
ARIMA(0,1,1)(2,1,0)_[12]_	1484.56	0.62
ARIMA(2,1,0)(0,1,1)_[12]_	1478.89	0.44
ARIMA(2,1,0)(2,1,0)_[12]_	1475.78	0.39
ARIMA(2,1,1)(0,1,1)_[12]_	1472.15	0.62
ARIMA(2,1,1)(1,1,0)_[12]_	1472.00	0.66
ARIMA(2,1,1)(2,1,0)_[12]_	1469.30	0.56
ARIMA(2,1,2)(0,1,0)_[12]_	1495.81	0.89

* Represents AIC minimum.

**Table 4 ijerph-19-06625-t004:** ARIMAX models that passed parameter tests and model diagnosis.

Model	Xreg	Lag	Β	T	*P*	AIC	Box–Ljung Test *p*
ARIMA(1,1,2)(2,1,0)_[12]_	/	/	/	/	/	1468.68	0.83
ARIMAX(1,1,2)(2,1,0)_[12]_	P	10	0.46	1.97	0.03	1328.84 *	0.94
ARIMAX(1,1,2)(2,1,0)_[12]_	T	9	20.37	1.66	0.04	1343.56	0.82
ARIMAX(1,1,2)(2,1,0)_[12]_	H	9	−6.10	1.74	0.04	1343.24	0.88
ARIMAX(1,1,2)(2,1,0)_[12]_	W	7	207.94	1.83	0.03	1368.46	0.76

* Represents AIC minimum.

**Table 5 ijerph-19-06625-t005:** MAE and RMSE of prediction by ARIMA and ARIMAX models.

Date	Actual	Predict		MAE		RMSE	
		Model 1	Model 2	Model 1	Model 2	Model 1	Model 2
2019.01	302	542	525	137.50	137.00	180.55	173.02
2019.02	201	188	520				
2019.03	521	511	541				
2019.04	880	879	577				
2019.05	806	1227	610				
2019.06	746	993	627				
2019.07	459	557	597				
2019.08	354	277	541				
2019.09	581	398	567				
2019.10	539	384	627				
2019.11	579	454	548				
2019.12	525	605	531				

## Data Availability

In this study, meteorological data for Chongqing from 2009 to 2019 were obtained from the *Chongqing Statistical Yearbook* (http://tjj.cq.gov.cn/ (accessed on 13 September 2021)).

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
