# Peer review of "Association between Meteorological Factors and Mumps and Models for Prediction in Chongqing, China"

_ijerph, 2022, doi:10.3390/ijerph19116625_

Round 1
Reviewer 1 Report
The manuscript entitled "Association between meteorological factors and prevalence of mumps and models for prediction in Chongqing, China" presents an interesting exercise on prediction of mumps prevalence. However, the most important drawback is lack of vaccination data, which could be a potential confouder for variations in the observed trends. In fact, the authors recognize that "It can be seen that the association between meteorological factors and mumps is not consistent in different country, and we doubt that these differences may be attributed to environmental and climatic characteristics in different regions." (line 229-232). What if this difference is associated with vaccination coverage? The authors need to consider this issue.
Other minor considerations:
- The manuscript has some linguistic inconsistencies.
- The methods section has to be expanded. The process of ARIMAX modeling has to described in more details.
- The subsection entitled "Data preprocessing" (lines 86-90) has to be moved to the Methods section.
- In Coblusion section the authors speculate that "It can provide a theoretical basis for relevant departments to control the mumps epidemic." (lines 251-252). How this could be done? By changing the climate and meteorological conditions? Is it relevant to conclude about prevention if the entire manuscript does not touch this issue?
Reviewer 2 Report
The revision is included in the attached document.

Reviewer 3 Report
see gramatical mistakes and add statistical analysis
Round 2
Reviewer 1 Report
None
Reviewer 2 Report
The authors have addressed successfully my comments and the paper can be accepted.